# Integration of machine learning XGBoost and SHAP models for NBA game outcome prediction and quantitative analysis methodology

**Yan Ouyang**[1,2☯], **Xuewei Li**[1,3,4☯], **Wenjia Zhou**[1,2☯], **Wei Hong**[1,2], **Weitao Zheng**[1,2,3,4], **Feng Qi**[1], **Liming Peng**[2,5]*

1 School of Intelligent Sports Engineering, Wuhan Sports University, Wuhan, Hubei, People's Republic of China, 2 Key Laboratory of Sports Engineering of General Administration of Sport of China, Wuhan Sports University, Wuhan, Hubei, People's Republic of China, 3 Engineering Research Center of Sports Health Intelligent Equipment of Hubei Province, Wuhan Sports University, Wuhan, Hubei, People's Republic of China, 4 Hubei Provincial Engineering and Technology Research Center of Sports Equipment, Wuhan Sports University, Wuhan, Hubei, People's Republic of China, 5 The College of Sports Science and Technology of Wuhan Sports University, Wuhan, Hubei, People's Republic of China

☯ These authors contributed equally to this work.
* wtplm@163.com

**Data Availability Statement:** All the data can be found at https://www.basketball-reference.com/. For more detailed instructions, please see "Data acquisition" in the Data processing procedures and

## Abstract

This study investigated the application of artificial intelligence in real-time prediction of professional basketball games, identifying the variations within performance indicators that are critical in determining the outcomes of the games. Utilizing games data from the NBA seasons 2021 to 2023 as the sample, the study constructed a real-time predictive model for NBA game outcomes, integrating the machine learning XGBoost and SHAP algorithms. The model simulated the prediction of game outcomes at different time of games and effectively quantified the analysis of key factors that influenced game outcomes. The study's results demonstrated that the XGBoost algorithm was highly effective in predicting NBA game outcomes. Key performance indicators such as field goal percentage, defensive rebounds, and turnovers were consistently related to the outcomes at all times during the game. In the first half of the game, assists were a key indicator affecting the outcome of the game. In the second half of the games, offensive rebounds and three-point shooting percentage were key indicators affecting the outcome of the games. The performance of the real-time prediction model for NBA game outcomes, which integrates machine learning XGBoost and SHAP algorithms, is found to be excellent and highly interpretable. By quantifying the factors that determine victory, it is able to provide significant decision support for coaches in arranging tactical strategies on the court. Moreover, the study provides reliable data references for sports bettors, athletes, club managers, and sponsors.

experimental results section.You can access the code using the following link: https://github.com/YanOuyang514/NBA-game-outcome-prediction-and-quantitative-analysis-methodology.git.

**Funding:** This research was funded by the 14th Five-Year-Plan Advantageous and Characteristic Disciplines (Groups) of Colleges and Universities in Hubei Province (Grant Number: E Jiao Yan No. [2021] 5). The funder (Liming Peng) participated in the revision of the manuscript and in the decision to publish. Hubei Provincial Social Science Fund General Project "Research on Personalized Recommendation of Online Sports Education Resources Based on Knowledge Graph" (Grant Number: 2021330), and the Scientific and Technological Research Project of Hubei Provincial Education Department (Grant Number: B2021189). The funder (Xuewei Li) had participated in study design, data collection, and analysis, the decision to publish, as well as preparation of the manuscript.

**Competing interests:** The authors have declared that no competing interests exist.

# 1 Introduction

With the rapid development of artificial intelligence and computer technology, "data-driven sports training and athletic decision-making" have now become a thriving focus of scholarly inquiry in competitive sports development. Scientific and accurate prediction of sports competitions and analysis of sports performance have also become increasingly important. Numerous studies have confirmed that machine learning techniques demonstrate significant potential in predicting game outcomes, analyzing sports performance, assessing physical fitness, and other related sports issues [1–9], making machine learning models a valuable tool for sports performance analysts.

While machine learning methods are powerful due to the complexity of their models, they are still limited by the difficulty of direct interpretation, known as the so-called "black box" issue. Currently, simple models such as logistic regression and decision trees are widely applied due to their ease of understanding and strong interpretability. However, they often fall short in achieving high predictive accuracy. To overcome the issues of these "black boxes", the Shapley Additive exPlanations (SHAP) method is utilized to interpret machine learning models and visualize individual variable predictions [10].

This research is designed to construct a real-time model for predicting the winning team in basketball games by integrating machine learning algorithms and to perform a quantitative analysis of the key factors affecting the game outcomes. The innovation in our research methodology lies in the use of a grouping strategy to build real-time game prediction models, leveraging the SHAP algorithm to interpret the optimal model, and conducting a quantitative analysis of the winning factors at different time of basketball games. This provides significant decision support for coaches' on-court dynamic decision-making and offers reliable data references for sports bettors, athletes, club managers, and sponsors.

Following the introduction, Section 2 thoroughly reviews the existing machine learning research in the field of sports, with a particular focus on advancements in basketball, providing a solid foundation for this study. Subsequently, Section 3 details the designed solution and the essential key technologies required for its implementation. Section 4 elucidates the data processing procedures and presents the experimental results. Section 5 delves into the discussion and analysis of the key factors determining victory or defeat at different time of the game. Section 6 demonstrates the practical application value of the predictive model through case studies. Finally, Section 7 summarizes the research findings and considers the potential limitations of the study.

# 2 Related studies

With the rapid development of artificial intelligence technology, its application in assisting sports event analysis and decision-making has garnered increasing attention, especially in ball games such as basketball, soccer, and American football. The feasibility and benefits of using various machine learning algorithms to predict game outcomes have been extensively explored in numerous studies [11–19]. For instance, Rodrigues and Pinto [11] (2022) employed machine learning algorithms such as Naive Bayes, K-Nearest Neighbors (KNN), Random Forest, Support Vector Machines (SVM), and Artificial Neural Networks (ANN) to analyze game data from five seasons of the English Premier League in order to construct a model for predicting game outcomes. By comparing the prediction accuracy of various models, they selected the one with the most superior performance and successfully validated the model's economic benefits using bookmaking data. Ötting [12] (2021) utilized Hidden Markov Models for predicting National Football League games, employing the 2018 National Football League season data as a sample, achieving an accuracy rate of 71.6%. Yaseen et al. [13] (2022) employed Logistic

Regression and SVM algorithms to analyze Major League Baseball (MLB) game data spanning the past 20 years, aiming to predict which teams would make it to the playoffs in 2019, with an accuracy rate of 77%.

Çene [20] (2022) explored the performance of seven different machine learning algorithms in predicting European league games from the 2016–2017 to the 2020–2021 seasons. The findings indicated that logistic regression, SVM, and ANN were the most effective models, with an overall accuracy rate of approximately 84%.

Cai et al. [21] (2019) constructed a hybrid ensemble learning framework based on the SVM using data from the Chinese Basketball Association (CBA) teams, achieving a prediction accuracy rate of 84%. Kaur and Jain [22] (2017) developed a hybrid fuzzy-SVM model (HFSVM) using regular season game data from 2015–2016 to predict NBA game outcomes. Comparative analysis revealed that this model outperformed the standard SVM model, with a prediction accuracy rate of 88.26%.

Moreover, Pai et al. [23] (2017) employed a hybrid model that combines SVM algorithms with decision tree algorithms to predict NBA games. Their analysis revealed that the hybrid model leverages the unique strengths of SVM and decision trees in generating rules and predicting game outcomes, providing insights to assist coaches in devising strategies. Huang and Lin [24] (2020) and Shi and Song [25] (2021) utilized regression trees and finite state Markov chain models, respectively, to predict NBA games and, based on NBA season data, validated the models' positive economic benefits in the betting market.

Zhao et al. [26] (2023) applied Graph Neural Networks (GNN) to predict basketball game outcomes by transforming structured data into unstructured graphs to reveal the complex passing interactions among players. By integrating machine learning algorithms with Graph Convolutional Network (GCN) models, they achieved a prediction accuracy rate of 71.54%, offering a novel perspective on team interactions.

Osken and Onay [27] (2022) utilized K-means and C-means clustering analysis to identify style types of NBA basketball players and constructed an ANN to predict NBA game outcomes based on player style types and game data. During the 2012–2018 NBA seasons, this method achieved a prediction accuracy rate of 76%.

In addition to predicting the outcomes of games, many scholars have applied logistic regression and decision tree algorithms to study the factors contributing to success in basketball. For instance, Leicht et al. [28] (2017) constructed logistic regression and Conditional Inference (CI) decision tree algorithms using data from men's Olympic games between 2004 and 2016 to predict game outcomes. They found that the logistic regression prediction model had a higher accuracy rate than the CI decision tree model. However, the CI decision tree offered greater practicality for coaches by addressing non-linear phenomena. The combination of points per game, defensive rebounds, turnovers, and steals was found to effectively explain the results of games, providing important guidance for the formulation of training and game strategies, thereby increasing the probability of success in men's Olympic basketball competitions.

Thabtah et al. [29] (2019) utilized Naive Bayes, ANN and Decision tree algorithms to model the historical data of NBA Finals from 1980 to 2017 in the Kaggle dataset. They discovered that the most effective method was a decision tree based on logistic regression, achieving an accuracy rate of 83%. Through feature selection analysis, it was found that defensive rebounds (DRB) was the most significant feature affecting NBA game outcomes. Additionally, three-point percentage (TPP), free throw percentage (FT), field goal percentage (FGP), and total rebounds (TRB) were identified as important factors.

## 3 Applied algorithms and methods

Building on the insights from related studies, our research introduces an innovative real-time prediction method for NBA game outcomes. The research flowchart is shown in Fig 1. The method integrates machine learning XGBoost [30] with SHAP [10] models. Considering the practical significance of the prediction model, we designed two different approaches during the model construction process: a real-time prediction model based on the technical performance indicators from the first two quarters and the first three quarters of the game, and a post-game prediction model constructed based on the full-game technical performance indicators.

To optimize prediction accuracy, this study employed methods such as Bayesian optimization and grid search to fine-tune the hyperparameters of seven mainstream machine learning algorithm models, including KNN, LightGBM, SVM, Random Forest, Logistic Regression, and Decision Tree. The performance of various metrics such as accuracy, precision, recall, F1 Score, and AUC value was compared in a ten-fold cross-validation experiment, demonstrating the superiority of the XGBoost algorithm.

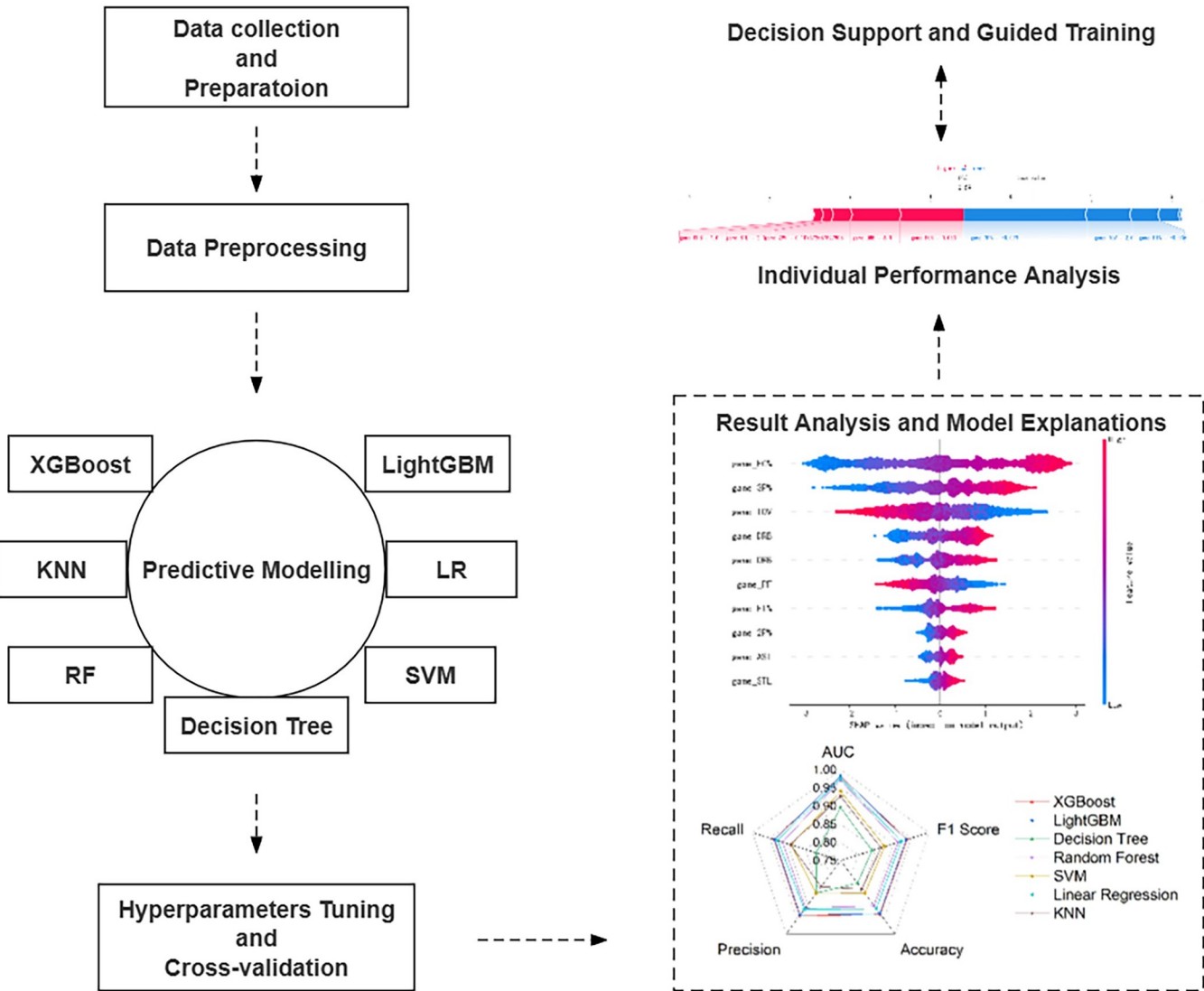

**Fig 1. NBA game outcome prediction model construction and application flowchart.**

Furthermore, the SHAP algorithm was introduced to enhance the interpretability of the XGBoost model's prediction process, quantifying the relationship between game technical performance indicators and the outcomes of games. As an example, the XGBoost model's application in predicting the outcomes of NBA games, analyzing the key factors for victory, and devising targeted training strategies was demonstrated using Game 2 of the 2023 NBA Finals.

## 3.1 XGBoost algorithm

XGBoost, short for "eXtreme Gradient Boosting," is an ensemble learning algorithm based on gradient boosting decision trees. Its core concept involves using decision trees as weak learners, iteratively fitting the residuals of the predictions from the previous iteration, ultimately forming a strong learner through weighted aggregation [30]. The basic formula for the model is presented in Eq (1):

$$\hat{y} = \sum_{k=1}^{K} f_k(x) \tag{1}$$

In Eq (1), "K" represents the total number of trees, and $f_k(x)$ denotes a specific decision tree. Each tree contributes to the final prediction by fitting the residuals of the previous iteration. The predicted output result after the t-th iteration is represented by Eq (2):

$$\hat{y}^{(t)} = \hat{y}^{(t-1)} + f_t(x) \tag{2}$$

Therefore, the objective function of XGBoost, which aims to minimize the difference between predicted and actual values, is represented by Eq (3):

$$\text{obj}(\theta) = \sum_{i=1}^{n} L(y_i, \hat{y}_i) + \sum_{k=1}^{K} \Omega(f_k) \tag{3}$$

The objective function consists of two parts: the loss function $L(y_i, \hat{y}_i)$ and the regularization term $\Omega(f_k)$. Here, $y_i$ represents the actual outcome of the game, and $\hat{y}_i$ is the predicted outcome from the model. The regularization term $\Omega(f_k)$ helps to control the complexity of the model, preventing overfitting.

According to the forward stage-wise training process, the structure of the first t-1 trees is constant. Therefore, the objective function after t iterations can be rewritten as shown in Eq (4):

$$\text{obj}(\theta)^{(t)} = \sum_{i=1}^{n} L(y_i, \hat{y}_i^{(t-1)} + f_t(x_i)) + \Omega(f_t) \tag{4}$$

To construct the optimal model by minimizing the objective function, a second-order Taylor expansion is used to approximate the loss function, leading to the updated objective function in Eq (5):

$$\text{obj}^{(t)} \approx \sum_{i=1}^{n} \left[ L(y_i, \hat{y}_i^{(t-1)}) + g_i f_t(x_i) + \frac{1}{2} h_i f_t^2(x_i) \right] + \Omega(f_t) \tag{5}$$

Here, $g_i$ and $h_i$ represent the first and second-order derivatives of the loss function, respectively. The complexity of the tree model, denoted by $\Omega(f_t)$, is represented in Eq (6):

$$\Omega(f_t) = \gamma T + \frac{1}{2} \lambda \sum_{j=1}^{T} \omega_j^2 \tag{6}$$

Where $T$ denotes the number of leaf nodes, $\gamma$ indicates the penalty for the number of leaf nodes, and $\lambda$ represents the L2 regularization term. The optimal weight for each leaf node $\omega_j$ can be derived, and thus the final optimized objective function is represented by Eq (7):

$$\text{obj}^{(t)} = \sum_{i=1}^{n} \left[ L(y_i, \hat{y}_i^{(t-1)}) + g_i f_t^*(x_i) + \frac{1}{2} h_i (f_t^*(x_i))^2 \right] + \Omega(f_t^*) \tag{7}$$

By iterating this process, XGBoost constructs an ensemble of decision trees that effectively minimizes the prediction error while maintaining model interpretability and complexity control.

## 3.2 SHAP algorithm

Machine learning algorithms have demonstrated excellent performance in predicting the outcomes of NBA games, but they also face the issue of being "interpretable". We often cannot comprehend the decision-making process of machine learning algorithms, which is commonly referred to as the "black box" model. In 2017, Lundberg and Lee [10] (2017) proposed an additive SHAP interpretation algorithm, inspired by game theory. The purpose of this algorithm is to calculate the Shap Values for each feature, reflecting the contribution of different features to the model. This allows for the interpretation of the prediction mechanism of complex machine learning algorithm models from both local and global perspectives [10,31]. The framework of the SHAP algorithm is shown in Fig 2.

SHAP interprets the predictions of a machine learning model as the sum of Shap Values for each input feature, as seen in Eq (8).

$$\hat{y} = f_0 + f(x_{i1}) + f(x_{i2}) + \ldots + f(x_{ij}) \tag{8}$$

In the equation, $\hat{y}$ represents the predicted value of the model. $f_0$ is the average predicted value over all training data of the model, which is referred to as the base values. $f(x_{ij})$ is the

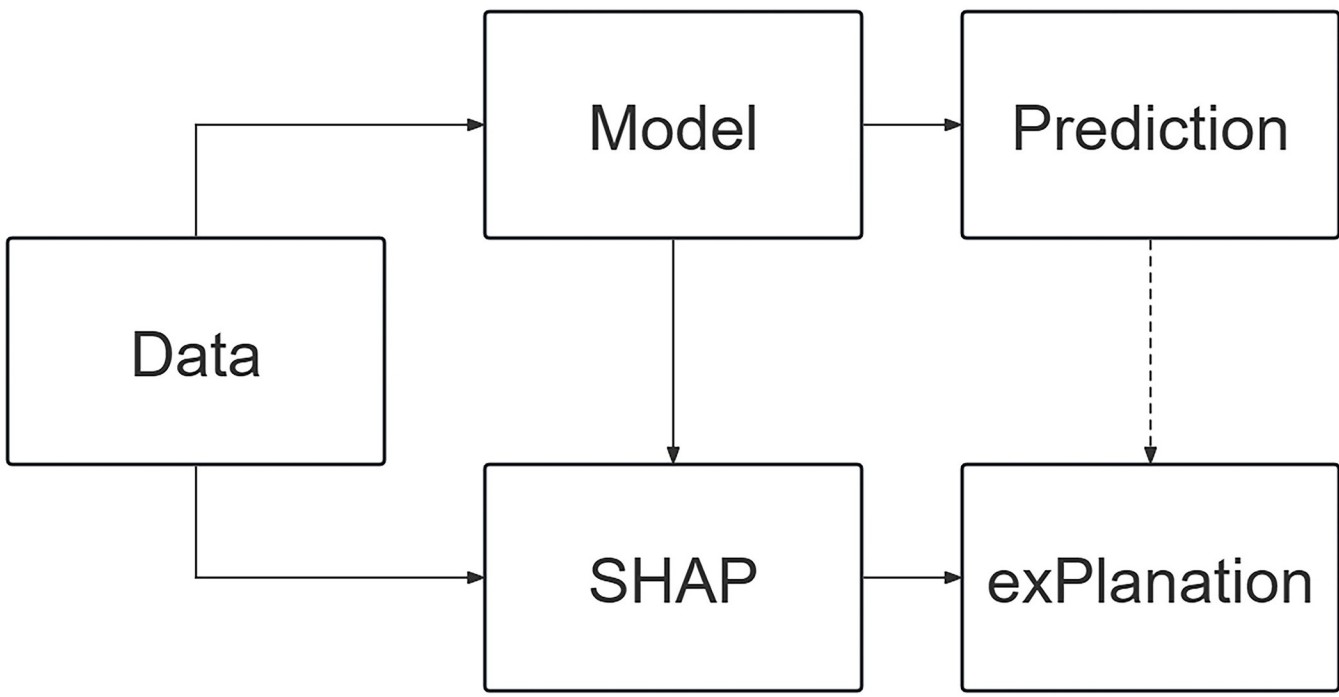

**Fig 2. Framework diagram of the SHAP algorithm.**

SHAP value for each feature of each sample, which reflects the positive or negative influence of each feature on the model's predicted outcome.

"Feature importance" as a traditional method calculates the significance of each feature by altering one feature variable while holding all other feature variables constant. However, this method fails to reflect the relationship between features and the model's predictive outcomes. In contrast, SHAP is capable of calculating the contribution values of features and visualizing the results, which allows for a direct understanding of the positive or negative impact of input features on the predicted outcomes of games. This, in turn, increases the explainability of the predictive model.

## 4 Data processing procedures and experimental results

### 4.1 Data processing procedures

**4.1.1 Data acquisition.** This study utilized a Python web scraper to collect relevant data on over 3,800 NBA games from the 2020–2021 season to the 2022–2023 season, across three seasons, from the Basketball Reference website (https://www.basketball-reference.com/). After removing outliers, including data points from preseason games, All-Star games, and invalid erroneous entries, a final set of 3,710 valid data points was obtained. The data includes various indicators such as basic game information, team-specific technical statistics for individual quarters, and team-wide technical statistics for the entire game. The study does not involve human or animal subjects and does not include any personal or sensitive information. Therefore, approval from an Institutional Review Board (IRB) or ethics committee was not required [32].

**4.1.2 Reliability and validity of data.** To validate the effectiveness of the data, a subset of 15 games (with at least five games from each season) was randomly selected. Two experienced basketball players (first-class athletes from China) reviewed the game footage and compared it with the data collected from the website. The results demonstrate a high level of data reliability, with an intraclass correlation coefficient (ICC) of 0.98.

**4.1.3 Data preparation.** The game outcome prediction problem was transformed into a binary classification issue. The target value result represented the binary classification label for the home team's win or loss, with a home team win/loss converted to the numerical values 1/0, respectively. Additionally, the data from the first two quarters and the first three quarters of the game were summed up to create new feature variables, with the prefixes H2, H3, and game added to distinguish them. Definitions of these variables can be found in Table 1. Leveraging a thorough understanding of the sport of basketball and the specifics of game outcome prediction, and considering the characteristics of the data indicators, an approach was adopted where features representing identical technical indicators for both the home and away teams are subtracted from each other. This method helped to mitigate situations where the technical statistics of the home and away teams were closely matched, reducing the interference of redundant information. Concurrently, it served to lower the dimensionality of the data, thereby enhancing the performance and efficiency of the predictive model.

Exploratory analysis was conducted by plotting a heatmap of feature correlations. The color of the heatmap indicates the correlation between two features: darker colors represent stronger positive correlations, while lighter colors represent stronger negative correlations. The numerical values represent the correlation coefficients between corresponding features. The asterisks reflect the significance levels of the correlation coefficients: no asterisk denotes $p > 0.05$, one asterisk denotes $0.01 < p < 0.05$, two asterisks denote $0.001 < p < 0.01$, and three asterisks denote $p < 0.001$. Taking the full-game technical statistics as an example, the correlation heatmap is shown in Fig 3.

**Table 1. Definition of selected technical game performance-related variables.**

| Indicators | Descriptions |
|---|---|
| H2_FG | Difference in field goals made from the first two quarters |
| H2_FGA | Difference in field goals attempted from the first two quarters |
| H2_FG% | Difference in field goal percentage from the first two quarters |
| H2_2P | Difference in two-point field goals made from the first two quarters |
| H2_2PA | Difference in two-point field goals attempted from the first two quarters |
| H2_2P% | Difference in two-point field goal percentage from the first two quarters |
| H2_3P | Difference in three-point field goals made from the first two quarters |
| H2_3PA | Difference in three-point field goals attempted from the first two quarters |
| H2_3P% | Difference in three-point field goal percentage from the first two quarters |
| H2_FT | Difference in free throws made from the first two quarters |
| H2_FTA | Difference in free throws attempted from the first two quarters |
| H2_FT% | Difference in free throw percentage from the first two quarters |
| H2_ORB | Difference in offensive rebounds from the first two quarters |
| H2_DRB | Difference in defensive rebounds from the first two quarters |
| H2_TRB | Difference in total number of rebounds from the first two quarters |
| H2_AST | Difference in assists from the first two quarters |
| H2_PF | Difference in personal fouls from the first two quarters |
| H2_STL | Difference in steals from the first two quarters |
| H2_TOV | Difference in turnovers from the first two quarters |
| H2_BLK | Difference in blocks from the first two quarters |
| H3_FG | Difference in field goals made from the first three quarters |
| H3_FGA | Difference in field goals attempted from the first three quarters |
| H3_FG% | Difference in field goal percentage from the first three quarters |
| H3_2P | Difference in two-point field goals made from the first three quarters |
| H3_2PA | Difference in two-point field goals attempted from the first three quarters |
| H3_2P% | Difference in two-point field goal percentage from the first three quarters |
| H3_3P | Difference in three-point field goals made from the first three quarters |
| H3_3PA | Difference in three-point field goals attempted from the first three quarters |
| H3_3P% | Difference in three-point field goal percentage from the first three quarters |
| H3_FT | Difference in free throws made from the first three quarters |
| H3_FTA | Difference in free throws attempted from the first three quarters |
| H3_FT% | Difference in free throw percentage from the first three quarters |
| H3_ORB | Difference in offensive rebounds from the first three quarters |
| H3_DRB | Difference in defensive rebounds from the first three quarters |
| H3_TRB | Difference in total number of rebounds from the first three quarters |
| H3_AST | Difference in assists from the first three quarters |
| H3_PF | Difference in personal fouls from the first three quarters |
| H3_STL | Difference in steals from the first three quarters |
| H3_TOV | Difference in turnovers from the first three quarters |
| H3_BLK | Difference in blocks from the first three quarters |
| game_FG | Difference in field goals made from the full game |
| game_FGA | Difference in field goals attempted from the full game |
| game_FG% | Difference in field goal percentage from the full game |
| game_2P | Difference in two-point field goals made from the full game |
| game_2PA | Difference in two-point field goals attempted from the full game |
| game_2P% | Difference in two-point field goal percentage from the full game |
| game_3P | Difference in three-point field goals made from the full game |

(*Continued*)

**Table 1.** (Continued)

| Indicators | Descriptions |
|---|---|
| game_3PA | Difference in three-point field goals attempted from the full game |
| game_3P% | Difference in three-point field goal percentage from the full game |
| game_FT | Difference in free throws made from the full game |
| game_FTA | Difference in free throws attempted from the full game |
| game_FT% | Difference in free throw percentage from the full game |
| game_ORB | Difference in offensive rebounds from the full game |
| game_DRB | Difference in defensive rebounds from the full game |
| game_TRB | Difference in total number of rebounds from the full game |
| game_AST | Difference in assists from the full game |
| game_PF | Difference in personal fouls from the full game |
| game_STL | Difference in steals from the full game |
| game_TOV | Difference in turnovers from the full game |
| game_BLK | Difference in blocks from the full game |
| Result | Home team win-loss result |

*Note.* The difference in scores between the home and away teams was calculated, with the home team's values subtracted from those of the away team.

We observed that there are strong correlations (correlation coefficients $\geq 0.8$) among variables such as game_FG and game_FG%, game_3PA and game_3P%, game_FTA and game_FT, and game_TRB and game_DRB, all with p-values less than 0.001. Consequently, we conducted an exploratory analysis of the full-game technical statistics dataset and plotted the corresponding scatter plots.

Analysis of the scatter plots (Fig 4) shows that the number of attempts in two-point shots, three-point shots, and free throws is linearly and positively correlated with the number of successful shots. Additionally, the shooting accuracy improves with an increase in the number of successful shots. Furthermore, there is a linear positive correlation between defensive rebounds and total rebounds, while offensive rebounds tend to decrease as defensive rebounds increase.

To avoid the high correlation between feature variables affecting the model's predictive performance, feature selection was conducted using a heatmap of feature correlations. The total rebounds were calculated as the sum of offensive and defensive rebounds, and the field goal percentage was determined as the ratio of field goal attempts to field goals made. Since the field goal percentage more accurately reflects the team's strength and condition, features such as field goal makes, field goal attempts, two-pointer makes, two-pointer attempts, three-pointer makes, three-pointer attempts, free throw makes, free throw attempts, and total rebounds were removed. Additionally, irrelevant features from the basic game information were also eliminated, constructing a new set of features for the sample dataset. The describe function was used to perform descriptive statistics on the NBA sample data, and the analysis results are listed in Table 2 below.

To ensure the robustness of our predictive model and to enhance the reliability and interpretability of our research findings, we employed logistic regression to test the significance of key performance Variables. Tables 3–5 present the logistic regression analysis results for different periods of the game.

From the logistic regression results, it is evident that field goal percentage, two-point field goal percentage, three-point field goal percentage, free throw percentage, offensive rebounds,

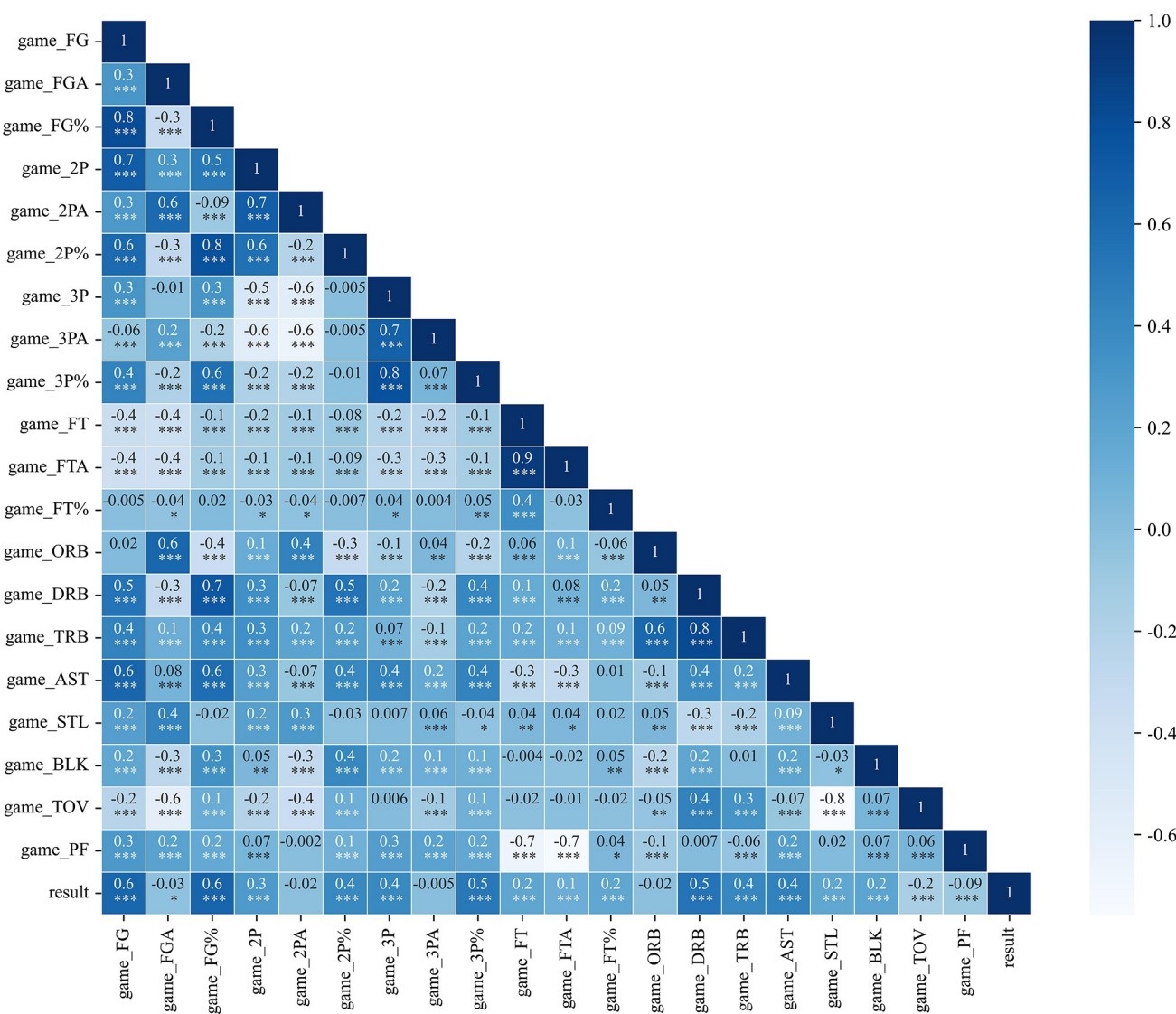

**Fig 3. Heatmap of NBA game technical statistics data.** The color of the heatmap indicates the correlation between two features: darker colors represent stronger positive correlations, while lighter colors represent stronger negative correlations. The numerical values represent the correlation coefficients between corresponding features. The asterisks reflect the significance levels of the correlation coefficients: no asterisk denotes $p > 0.05$, one asterisk denotes $0.01 < p < 0.05$, two asterisks denote $0.001 < p < 0.01$, and three asterisks denote $p < 0.001$.

defensive rebounds, assists, personal fouls, and turnovers significantly impact the game outcome across different game periods. In contrast, blocks and steals do not significantly affect the game outcome. Additionally, personal fouls and turnovers have a negative impact on the game outcome.

Given the crucial role of blocks and steals in influencing game results and their confirmed importance in previous research, we decided to retain all 11 indicators [28, 33–35], including blocks and steals, across the three distinct time period datasets.

## 4.2 Model training and experimental results

To enhance the practical significance and application value of the predictive model, this study grouped and merged the NBA game dataset according to the duration of the games,

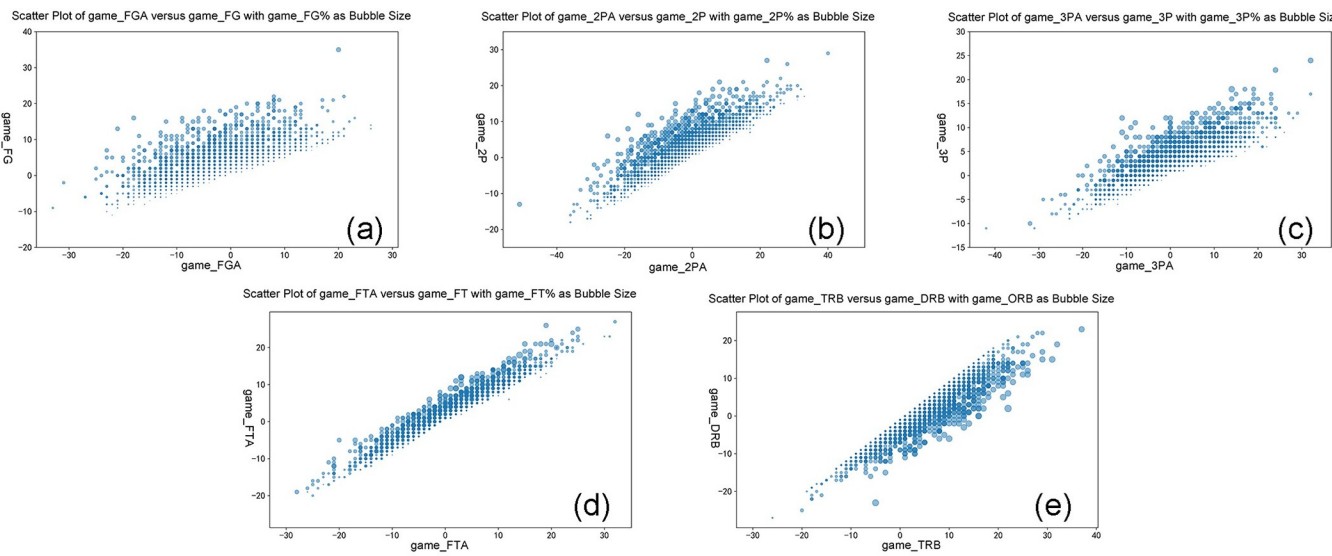

**Fig 4. Exploratory scatter plots of technical statistic relationships from full game.**

**Table 2. List of feature data for the sample.**

| Indicators | Mean | Std | Min | 0.25 | 0.5 | 0.75 | Max |
|---|---|---|---|---|---|---|---|
| H2_FG% | 0.008 | 0.105 | −0.373 | −0.062 | 0.010 | 0.078 | 0.373 |
| H2_2P% | 0.010 | 0.142 | −0.470 | −0.083 | 0.010 | 0.106 | 0.667 |
| H2_3P% | 0.006 | 0.169 | −0.701 | −0.108 | 0.008 | 0.119 | 0.579 |
| H2_FT% | 0.006 | 0.225 | −1 | −0.133 | 0 | 0.143 | 1 |
| H2_ORB | −0.038 | 3.543 | −12 | −2 | 0 | 2 | 11 |
| H2_DRB | 0.367 | 4.939 | −17 | −3 | 0 | 4 | 19 |
| H2_AST | 0.242 | 4.783 | −17 | −3 | 0 | 3 | 16 |
| H2_STL | −0.026 | 2.767 | −10 | −2 | 0 | 2 | 10 |
| H2_BLK | 0.129 | 2.236 | −9 | −1 | 0 | 2 | 10 |
| H2_TOV | −0.020 | 3.502 | −12 | −2 | 0 | 2 | 12 |
| H2_PF | −0.130 | 3.231 | −12 | −2 | 0 | 2 | 12 |
| H3_FG% | 0.009 | 0.087 | −0.364 | −0.052 | 0.009 | 0.066 | 0.308 |
| H3_2P% | 0.010 | 0.118 | −0.463 | −0.073 | 0.013 | 0.090 | 0.470 |
| H3_3P% | 0.008 | 0.137 | −0.505 | −0.083 | 0.010 | 0.099 | 0.514 |
| H3_FT% | 0.001 | 0.169 | −1.041 | −0.110 | 0.000 | 0.111 | 0.833 |
| H3_ORB | −0.020 | 4.511 | −17 | −3 | 0 | 3 | 16 |
| H3_DRB | 0.554 | 6.145 | −19 | −4 | 0 | 5 | 22 |
| H3_AST | 0.395 | 6.007 | −23 | −4 | 0 | 4 | 22 |
| H3_STL | −0.053 | 3.468 | −13 | −2 | 0 | 2 | 13 |
| H3_BLK | 0.178 | 2.781 | −14 | −2 | 0 | 2 | 12 |
| H3_TOV | −0.045 | 4.418 | −17 | −3 | 0 | 3 | 16 |
| H3_PF | −0.158 | 3.983 | −17 | −3 | 0 | 2 | 14 |
| game_FG% | 0.007 | 0.076 | −0.313 | −0.045 | 0.008 | 0.058 | 0.296 |
| game_2P% | 0.008 | 0.102 | −0.392 | −0.062 | 0.010 | 0.076 | 0.352 |
| game_3P% | 0.007 | 0.121 | −0.433 | −0.071 | 0.009 | 0.088 | 0.501 |
| game_FT% | 0.001 | 0.142 | −0.480 | −0.092 | 0.000 | 0.096 | 0.575 |
| game_ORB | 0.065 | 5.387 | −22 | −4 | 0 | 4 | 20 |

*(Continued)*

**Table 2.** (Continued)

| Indicators | Mean | Std | Min | 0.25 | 0.5 | 0.75 | Max |
|---|---|---|---|---|---|---|---|
| game_DRB | 0.691 | 7.317 | −27 | −4 | 1 | 6 | 25 |
| game_AST | 0.567 | 6.847 | −22 | −4 | 1 | 5 | 28 |
| game_STL | −0.054 | 4.040 | −13 | −3 | 0 | 3 | 17 |
| game_BLK | 0.184 | 3.317 | −17 | −2 | 0 | 2 | 15 |
| game_TOV | −0.046 | 5.074 | −21 | −3 | 0 | 3 | 17 |
| game_PF | −0.211 | 4.915 | −20 | −4 | 0 | 3 | 17 |
| Result | 0.554 | 0.497 | 0 | 0 | 1 | 1 | 1 |

*Note*. Std = standard deviation; Min = minimum; 0.25 = lower quartile; 0.5 = median; 0.75 = upper quartile; Max = maximum.

**Table 3. Logistic regression analysis results of key performance variables during the first two quarters.**

| Indicators | Estimated coeff.($\hat{\beta}$) | Standard Error ($\hat{SE}$) | z | *p*-value | 95% Confidence Interval | |
|---|---|---|---|---|---|---|
| H2_FG% | 3.0548 | 1.446 | 2.113 | 0.035 | 0.221 | 5.888 |
| H2_2P% | 3.0312 | 0.824 | 3.677 | 0.000 | 1.416 | 4.647 |
| H2_3P% | 2.8175 | 0.52 | 5.42 | 0.000 | 1.799 | 3.836 |
| H2_FT% | 0.7101 | 0.183 | 3.886 | 0.000 | 0.352 | 1.068 |
| H2_ORB | 0.1096 | 0.014 | 7.614 | 0.000 | 0.081 | 0.138 |
| H2_DRB | 0.0624 | 0.017 | 3.587 | 0.000 | 0.028 | 0.096 |
| H2_AST | 0.0285 | 0.011 | 2.61 | 0.009 | 0.007 | 0.050 |
| H2_STL | 0.011 | 0.022 | 0.506 | 0.613 | -0.031 | 0.053 |
| H2_BLK | 0.0118 | 0.019 | 0.63 | 0.529 | -0.025 | 0.048 |
| H2_TOV | -0.1506 | 0.02 | -7.501 | 0.000 | -0.190 | -0.111 |
| H2_PF | -0.059 | 0.013 | -4.655 | 0.000 | -0.084 | -0.034 |
| Constant | 0.1736 | 0.038 | 4.512 | 0.000 | 0.098 | 0.249 |

**Table 4. Logistic regression analysis results of key performance variables during the first three quarters.**

| Indicators | Estimated coeff. ($\hat{\beta}$) | Standard Error ($\hat{SE}$) | z | *p*-value | 95% Confidence Interval | |
|---|---|---|---|---|---|---|
| H3_FG% | 7.8625 | 2.042 | 3.85 | 0.000 | 3.860 | 11.865 |
| H3_2P% | 6.2073 | 1.199 | 5.178 | 0.000 | 3.858 | 8.557 |
| H3_3P% | 7.1168 | 0.775 | 9.187 | 0.000 | 5.598 | 8.635 |
| H3_FT% | 2.3081 | 0.283 | 8.15 | 0.000 | 1.753 | 2.863 |
| H3_ORB | 0.1542 | 0.014 | 11.077 | 0.000 | 0.127 | 0.181 |
| H3_DRB | 0.0678 | 0.016 | 4.244 | 0.000 | 0.036 | 0.099 |
| H3_AST | 0.023 | 0.01 | 2.307 | 0.021 | 0.003 | 0.043 |
| H3_STL | -0.0078 | 0.02 | -0.396 | 0.692 | -0.047 | 0.031 |
| H3_BLK | 0.0196 | 0.018 | 1.112 | 0.266 | -0.015 | 0.054 |
| H3_TOV | -0.2075 | 0.019 | -11.006 | 0.000 | -0.244 | -0.171 |
| H3_PF | -0.0819 | 0.012 | -6.857 | 0.000 | -0.105 | -0.058 |
| Constant | 0.1238 | 0.044 | 2.8 | 0.005 | 0.037 | 0.211 |

**Table 5. Logistic regression analysis results of key performance variables during the full game.**

| Indicators | Estimated coeff. ($\hat{\beta}$) | Standard Error ($\hat{SE}$) | z | p-value | 95% Confidence Interval | |
|---|---|---|---|---|---|---|
| game_FG% | 28.7421 | 4.427 | 6.492 | 0.000 | 20.065 | 37.419 |
| game_2P% | 31.7362 | 2.948 | 10.766 | 0.000 | 25.959 | 37.514 |
| game_3P% | 39.1816 | 2.436 | 16.085 | 0.000 | 34.407 | 43.956 |
| game_FT% | 10.4031 | 0.784 | 13.268 | 0.000 | 8.866 | 11.940 |
| game_ORB | 0.4597 | 0.03 | 15.305 | 0.000 | 0.401 | 0.519 |
| game_DRB | 0.1905 | 0.027 | 7.171 | 0.000 | 0.138 | 0.243 |
| game_AST | 0.0468 | 0.017 | 2.826 | 0.005 | 0.014 | 0.079 |
| game_STL | 0.0637 | 0.033 | 1.914 | 0.056 | -0.002 | 0.129 |
| game_BLK | 0.0084 | 0.027 | 0.311 | 0.756 | -0.044 | 0.061 |
| game_TOV | -0.627 | 0.041 | -15.319 | 0.000 | -0.707 | -0.547 |
| game_PF | -0.3056 | 0.022 | -13.792 | 0.000 | -0.349 | -0.262 |
| Constant | -0.0324 | 0.084 | -0.384 | 0.701 | -0.198 | 0.133 |

constructing datasets for the first two quarters, the first three quarters, and the full game to make real-time predictions of game outcomes at the end of the second and third quarters. An NBA game outcome prediction model was built based on the XGBoost algorithm. Hyperparameter tuning was conducted for the XGBoost algorithm and five other main-stream machine learning algorithms—KNN, LightGBM, SVM, Random Forest, Logistic Regression, and Decision Tree—using methods such as Bayesian optimization and grid search. The optimal predictive model architecture was obtained through a ten-fold cross-validation comparative experiment, combined with evaluation metrics. Finally, the SHAP algorithm was introduced for interpretability analysis of the best model, to uncover the key factors that determine the outcome of the games.

This study employs five classification performance metrics—AUC, F1 Score, accuracy, precision, and recall—to evaluate the quality of the NBA game outcome prediction model. When a model demonstrates good performance across these metrics in comparative experiments, it is considered to have superior predictive capabilities. Accuracy, precision, and recall respectively reflect the performance of a predictive model in "being right", "being precise", and "being comprehensive". The F1 Score combines precision and recall, reflecting the robustness of the model. The probabilistic meaning of the AUC value is the probability that, when randomly selecting a pair of positive and negative samples, the positive sample will have a higher score than the negative sample. The AUC value ranges from 0 to 1, with a higher AUC indicating a higher predictive value of the model.

The classification prediction results for game outcomes are presented in the confusion matrix as shown in Table 6. 1. Games where the actual result was a home team win and were predicted as a home team win are True Positives (TP); 2. Games where the actual result was a home team loss but were predicted as a home team win are False Positives (FP); 3. Games where the actual result was a home team loss and were predicted as a home team loss are True

**Table 6. Confusion matrix of classification results.**

| Home Team Win-Loss Result | Predicted Win | Predicted Loss |
|---|---|---|
| **Actual Win** | TP | FN |
| **Actual Loss** | FP | TN |

*Note.* TP = True Positives; FP = False Positives; TN = True Negatives; FN = False Negatives.

Negatives (TN); 4. Games where the actual result was a home team win but were predicted as a home team loss are False Negatives (FN).

Based on the confusion matrix, the accuracy, precision, recall, and F1 Score can be calculated using the following equations:

$$accuracy = \frac{TP + TN}{TP + FP + FN + TN} \tag{9}$$

$$precision = \frac{TP}{TP + FP} \tag{10}$$

$$recall = \frac{TP}{TP + FN} \tag{11}$$

$$F1\ Score = \frac{2 \times precision \times recall}{precision + recall} \tag{12}$$

**4.2.1 Ten-fold cross-validation comparative experiment.** After hyperparameter tuning using Bayesian optimization and grid search, the evaluation metric results of the predictive models for different time periods by various algorithms in the ten-fold cross-validation comparative experiments are listed in Tables 7–9 and shown in Fig 5: The XGBoost algorithm exhibits optimal performance in predicting the outcomes of NBA games. In terms of the AUC and F1 Score metrics, the XGBoost algorithm performed excellently, consistently ranking in the top 2 across the ten-fold cross-validation comparative experiments for the three different time periods. Regarding the accuracy and precision metrics, the XGBoost algorithm consistently showed the best performance. However, in terms of recall, the XGBoost algorithm ranked 4th, 3rd, and 1st in the comparative experiments for the three different time periods, with recall values of 0.775, 0.807, and 0.939, respectively.

In summary, across the ten-fold cross-validation experiments for three different periods (first two quarters, first three quarters, and the full game), the XGBoost algorithm outperformed other mainstream machine learning algorithms in the comprehensive evaluation of five performance metrics, demonstrating the most ideal predictive effect. It effectively captures the complex nonlinear relationship between NBA game technical indicators and game outcomes. The XGBoost algorithm model optimizes the objective function through a second-order Taylor expansion, enhancing computational accuracy and iterative efficiency.

**Table 7. Comparative results of performance evaluation metrics for NBA game outcome prediction models (first two quarters period).**

| Algorithm | AUC | F1 Score | Accuracy | Precision | Recall |
|---|---|---|---|---|---|
| XGBoost | 0.783 | 0.754 | **0.720** | **0.736** | 0.775 |
| LightGBM | 0.777 | **0.770** | 0.706 | 0.679 | **0.891** |
| Decision Tree | 0.722 | 0.735 | 0.687 | 0.691 | 0.786 |
| Random Forest | 0.757 | 0.739 | 0.699 | 0.712 | 0.768 |
| SVM | 0.779 | 0.749 | 0.708 | 0.715 | 0.787 |
| Linear Regression | **0.798** | 0.743 | 0.715 | 0.730 | 0.769 |
| KNN | 0.760 | 0.738 | 0.697 | 0.708 | 0.773 |

*Note*. Bolded text indicates the best result in each column, while underlined text indicates the second-best result in each column. SVM = Support Vector Machines; KNN = K-Nearest Neighbors.

**Table 8. Comparative results of performance evaluation metrics for NBA game outcome prediction models (first three quarters period).**

| Algorithm | AUC | F1 Score | Accuracy | Precision | Recall |
|---|---|---|---|---|---|
| **XGBoost** | 0.876 | **0.820** | **0.798** | **0.834** | 0.807 |
| **LightGBM** | **0.879** | 0.815 | 0.792 | 0.829 | 0.802 |
| **Decision Tree** | 0.773 | 0.734 | 0.721 | 0.773 | 0.700 |
| **Random Forest** | 0.859 | 0.796 | 0.773 | 0.785 | 0.810 |
| **SVM** | 0.845 | 0.784 | 0.759 | 0.771 | 0.798 |
| **Linear Regression** | 0.870 | 0.807 | 0.785 | 0.796 | **0.818** |
| **KNN** | 0.832 | 0.769 | 0.741 | 0.755 | 0.784 |

*Note*. Bolded text indicates the best result in each column, while underlined text indicates the second-best result in each column. SVM = Support Vector Machines; KNN = K-Nearest Neighbors.

Additionally, regularization terms are added to the loss function to control the model's complexity and prevent overfitting.

Overall, the LightGBM algorithm, an ensemble learning method, showed the closest performance to the XGBoost algorithm, reflecting the superiority of ensemble algorithms to some extent. Following these, the Logistic Regression algorithm performed slightly behind the LightGBM and XGBoost algorithms, which may be attributed to its sensitivity to multicollinearity among independent variables. The Decision Tree algorithm lagged behind the other machine learning algorithms in all metrics, possibly due to its difficulty in handling data with strong feature correlations. The research results indicate that, based on data preprocessing and feature extraction, the model presented in this study can better predict the outcomes of NBA games in real-time compared to existing studies.

**4.2.2 Analysis of factors influencing game outcomes at different time.** SHAP provides powerful and diverse data visualization charts to demonstrate the interpretability of the model. Based on the XGBoost real-time game outcome prediction model discussed earlier, SHAP quantifies and ranks the importance of features that influence the outcome of the games, as listed in Table 10. The SHAP feature summary Fig 6 illustrates that field goal percentage, defensive rebounds, and turnovers consistently rank in the top four for SHAP importance across different time of the game. Additionally, assists rank 3rd in importance during the first two quarters, but drop to 8th and 9th during the first three quarters and the full game, respectively. Offensive rebounds and three-point shooting percentages rank 9th and 8th in the first two quarters, respectively, and then rise to 5th and 3rd during the first three quarters and full

**Table 9. Comparative results of performance evaluation metrics for NBA Game outcome prediction models (full game period).**

| Algorithm | AUC | F1 Score | Accuracy | Precision | Recall |
|---|---|---|---|---|---|
| **XGBoost** | 0.982 | **0.939** | **0.933** | **0.938** | **0.939** |
| **LightGBM** | **0.983** | 0.936 | 0.930 | 0.935 | 0.937 |
| **Decision Tree** | 0.896 | 0.839 | 0.827 | 0.860 | 0.819 |
| **Random Forest** | 0.971 | 0.913 | 0.904 | 0.911 | 0.915 |
| **SVM** | 0.941 | 0.875 | 0.861 | 0.862 | 0.890 |
| **Linear Regression** | 0.976 | 0.922 | 0.914 | 0.916 | 0.929 |
| **KNN** | 0.927 | 0.865 | 0.847 | 0.839 | 0.892 |

*Note*. Bolded text indicates the best result in each column, while underlined text indicates the second-best result in each column. SVM = Support Vector Machines; KNN = K-Nearest Neighbors.

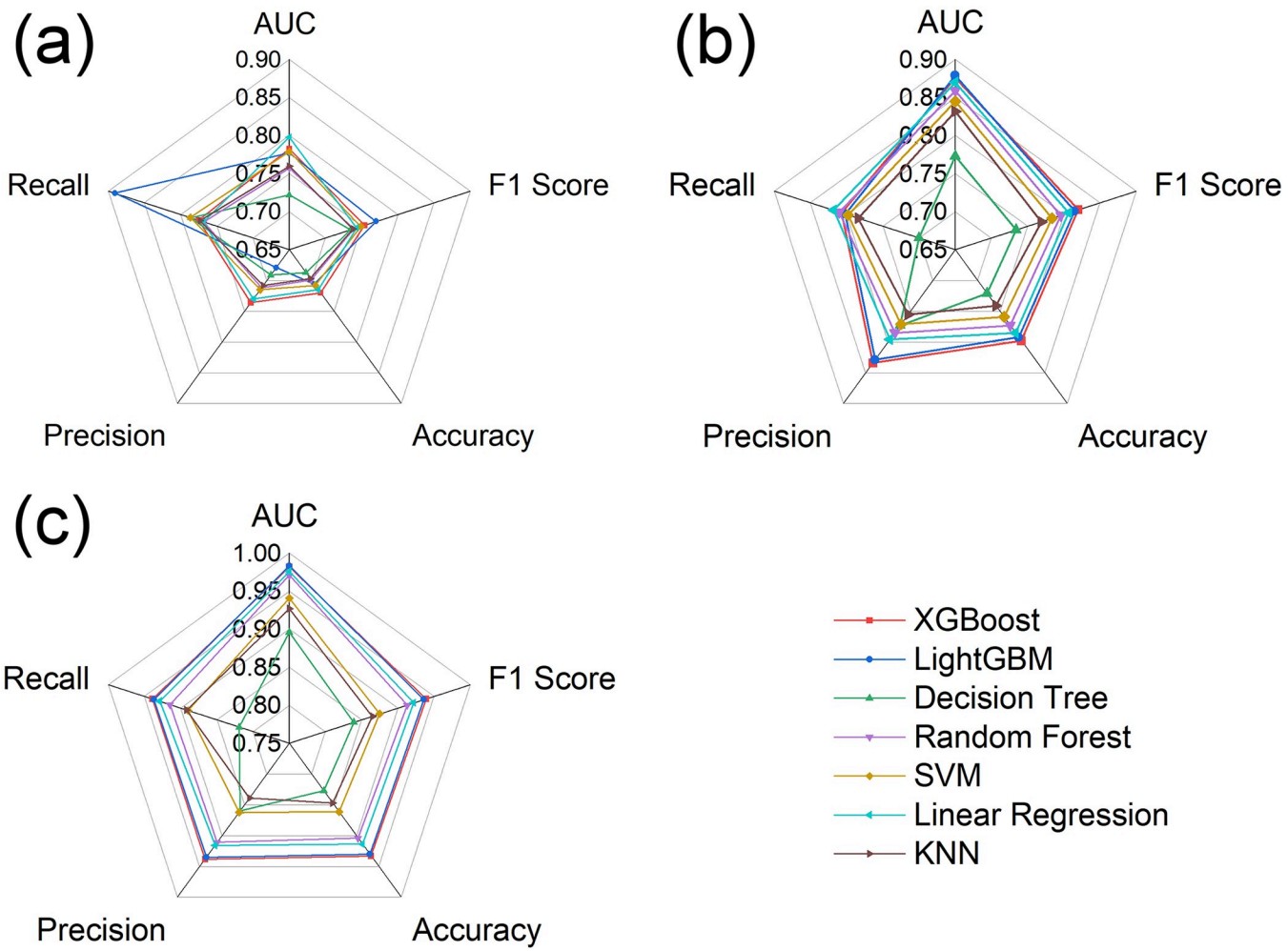

**Fig 5. Comparative chart of performance evaluation metrics for NBA game outcome prediction models.** (a) first two quarters period, (b) first three quarters period, (c) full game period. SVM = Support Vector Machines; KNN = K-Nearest Neighbors.

**Table 10. Comparison of SHAP feature importance at different time of the game.**

| Rank | First two quarters | | First three quarters | | Full game | |
|---|---|---|---|---|---|---|
| | Indicators | Values | Indicators | Values | Indicators | Values |
| 1 | H2_FG% | 0.142 | H3_FG% | 0.494 | game_FG% | 1.447 |
| 2 | H2_DRB | 0.060 | H3_TOV | 0.187 | game_3P% | 0.898 |
| 3 | H2_AST | 0.056 | H3_3P% | 0.165 | game_TOV | 0.790 |
| 4 | H2_TOV | 0.046 | H3_DRB | 0.162 | game_DRB | 0.555 |
| 5 | H2_FT% | 0.036 | H3_ORB | 0.133 | game_ORB | 0.459 |
| 6 | H2_PF | 0.035 | H3_FT% | 0.109 | game_PF | 0.447 |
| 7 | H2_STL | 0.033 | H3_PF | 0.097 | game_FT% | 0.391 |
| 8 | H2_3P% | 0.032 | H3_AST | 0.068 | game_2P% | 0.198 |
| 9 | H2_ORB | 0.026 | H3_STL | 0.065 | game_AST | 0.187 |
| 10 | H2_2P% | 0.019 | H3_2P% | 0.064 | game_STL | 0.149 |
| 11 | H2_BLK | 0.013 | H3_BLK | 0.014 | game_BLK | 0.015 |

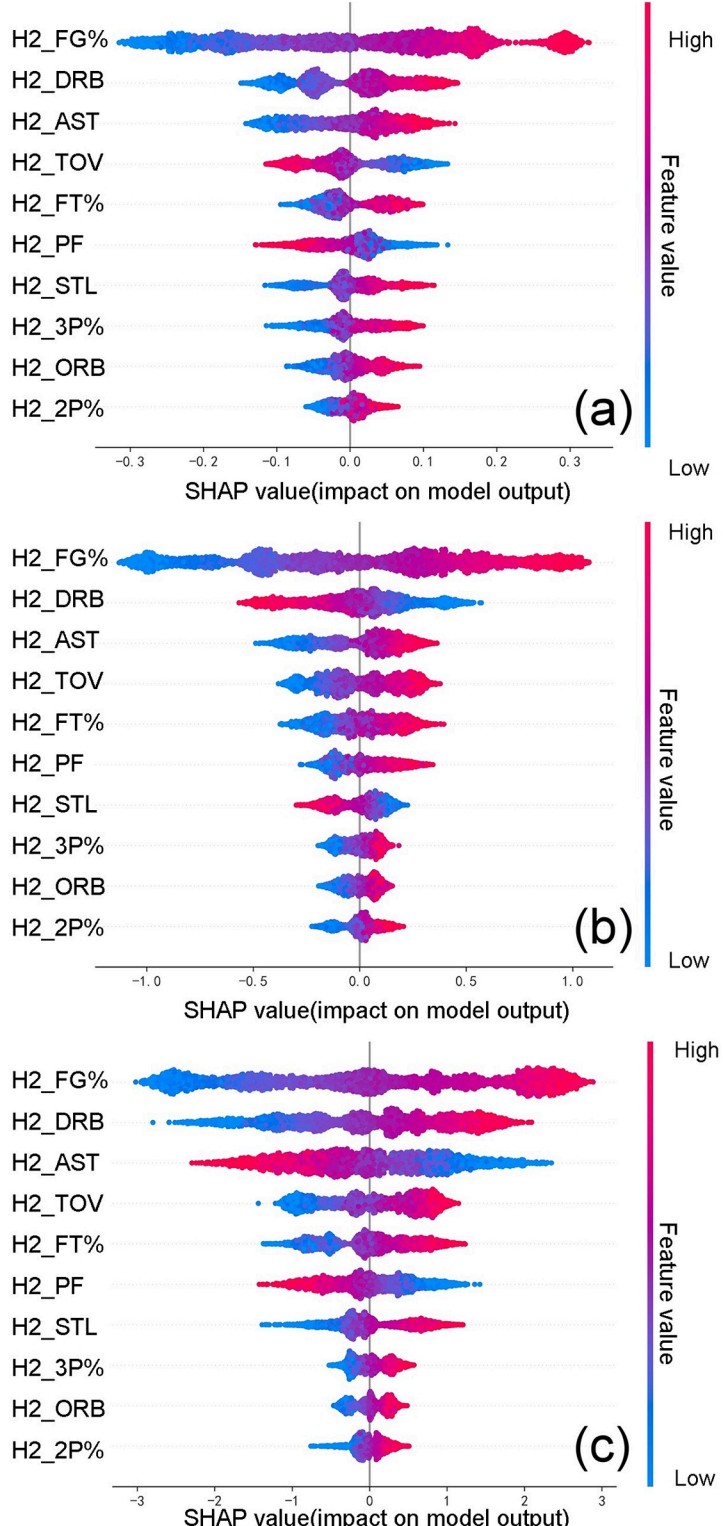

**Fig 6. Summary chart of SHAP feature importance at different time of the game.** (a) first two quarters period, (b) first three quarters period, (c) full game period.

game. Blocks, on the other hand, consistently rank 11th in SHAP importance across all time of the game.

## 5 Discussion

Team technical statistics are one of the essential tools for analyzing games. Delving deeply and exploring technical statistics from multiple angles can help us understand the current state of play and the offensive and defensive technical characteristics and applications of our opponents [36]. This study found that field goal percentage, defensive rebounds, and turnovers are key factors influencing the outcome of games across different periods.

The results indicated that Field goal percentage consistently holds the top spot in importance across three distinct periods within the game, corroborating prior studies that have emphasized the critical role of Field goal percentage in clinching victories in basketball contests [37–40]. Previously, Gómez et al. [41] (2008) and Ibáñez et al. [42] (2019) have indicated that the essence of basketball is striving to score more points than the opponent. A higher Field goal percentage can lead to a greater scoring advantage over the adversary, thus increasing the likelihood of winning the games and competitions.

Defensive rebounds have been proven to be a key winning factor in high-level basketball games [29, 37–39, 43]. Specifically, a defensive rebound occurs when a defensive player secures the basketball after an unsuccessful shot by the offensive team, thereby gaining possession. Defensive rebounds reflect the team's overall defensive rotation and coordination, the execution of defensive tactics, as well as individual players' one-on-one defensive abilities and attitudes. Sampaio et al. [44] (2010) have shown that defensive players should apply aggressive defense against their assigned opponents, forcing them into turnovers or contested shots, and actively fight for defensive rebounds. The protection of defensive rebounds is closely related to the rapid initiation of fast breaks; the direct purpose of contesting defensive rebounds is to transition from defense to offense, creating counterattacks and thus gaining opportunities to organize offensive plays and score. Additionally, contesting for defensive rebounds can limit the opponent's second-chance points and increase one's own offensive opportunities.

High-level basketball games are characterized by intense competition, and turnovers are an inevitable possibility. However, in basketball, turnovers signify a change of possession and a loss of offensive opportunities, directly impacting the game's scoring and the team's morale and confidence, thereby reducing the probability of winning the game [45]. A research by Leicht et al. [28] (2017) indicates that in World Cup competitions, strong teams may have certain advantages in terms of physical fitness and tactical skills. However, frequent turnovers by players during a particular stage can create psychological pressure on the team, disrupt the team's game rhythm and established tactics. If players continue to turnover the ball frequently, coaches should make timely decisions to substitute and rotate the lineup to ensure the effective implementation of the team's tactics. Only by minimizing the number of turnovers and enhancing the team's offensive stability can a team secure victory in the game.

Additionally, in the first two quarters of the game, assists are a key indicator affecting the outcome. In the latter two quarters, offensive rebounds and three-point shooting percentages become critical factors. This observation is consistent with the conclusions drawn by Jorge Malarranha, whose research underscores that offensive rebounds have a greater influence in the second half [39]. However, it is important to note that the decline in the importance ranking of assists in the latter two quarters does not mean that their impact on the game's outcome has diminished. Zheng [46] (2022) have shown that in the latter half of NBA games, especially during crucial moments, the number and rate of assists may decrease as individual offensive plays increase, turning assists into opportunities created by team coordination. As the game

nears its end, every offensive possession becomes vitally important. For the leading team, securing an offensive rebound not only provides an additional offensive chance but also prevents the opponent from scoring on a fast break, allowing the team to slow down the game and maintain their lead. For the trailing team, securing offensive rebounds is even more critical, as each offensive opportunity is key to closing or extending the score gap. Moreover, for teams looking to tie or pull away in the score during this period, the benefits of making three-pointers become significantly higher.

Therefore, during the preparation for international competitions, coaches need to start with the key details behind the data. While adhering to the general winning patterns, they should develop personalized training programs based on the significant factors influencing the game. This approach aims to address the players' weaknesses, diversify scoring methods, and enhance shooting stability. Additionally, coaches should devise tactics that game the individual playing styles of their players and ensure that these tactics are practiced to the point of seamless cooperation and ease of execution, thereby improving the players' abilities and the team's overall performance.

## 6 Match result prediction and application analysis

The 2022–2023 NBA Finals officially commenced on June 2, 2023, with the Miami Heat, ranked 8th in the Eastern Conference, making a remarkable underdog run to face the Denver Nuggets, the top-ranked team in the Western Conference. The Nuggets emerged victorious in the Finals with a 4–1 series win, securing their first-ever NBA championship title. Applying a real-time game outcome prediction model to the analysis of decisive factors in actual games provided significant practical value for team game analysis, tactical strategy arrangement, and the coaches' dynamic decision-making on the court.

In Game 2 of the NBA Finals, the Denver Nuggets, playing at home, fell short against the Miami Heat with a score of 108 to 111, marking the Heat's sole victory in the Finals series. Taking Game 2 as an example, SHAP feature analysis effect diagrams for different time periods was created, as Figs 7–9 demonstrate. The base value represents the average predicted probability of win or loss for the training set of games, with red areas indicating that the feature has a positive effect on the prediction outcome, and blue areas indicating a negative effect. The length of the color bars reflects the magnitude of the impact, with longer bars signifying a greater influence of that particular feature on the prediction result. In conjunction with the feature engineering of this study and the actual game situation, it can be deduced that the technical statistical indicators for the sample are the differences in technical statistics between the Nuggets and the Heat, with the prediction target being the probability of the Nuggets winning Game 2.

As shown in Fig 7 and Table 11, by the end of the second quarter of Game 2, the top four features impacting the game result, in order of Shap Values, were H2_FG%, H2_AST, H2_PF,

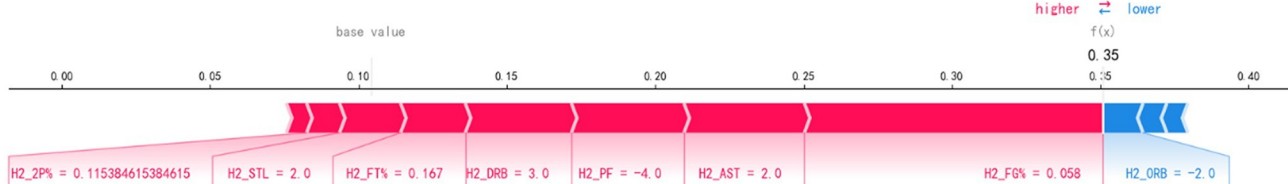

**Fig 7. Interpretation of SHAP force plot: analysis of performance in the first two quarters of G2 game.** The base value represents the average predicted probability of win or loss for the sample set of games, with red areas indicating that the feature has a positive effect on the prediction outcome, and blue areas indicating a negative effect. The length of the color bars reflects the magnitude of the impact, with longer bars signifying a greater influence of that particular feature on the prediction result. Features and their sample values are indicated below the color bars. Footnotes for Figs 8 and 9 are identical to that of Fig 7.

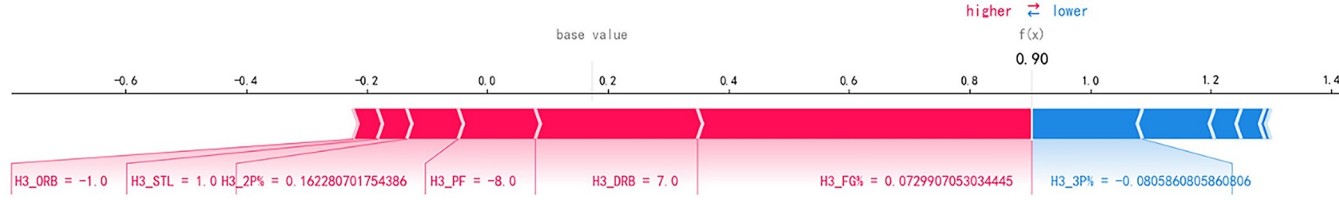

**Fig 8. Interpretation of SHAP force plot: analysis of performance in the first three quarters of G2 game.**

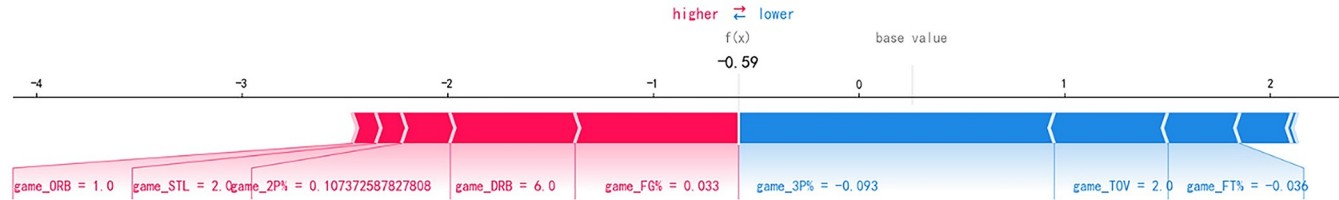

**Fig 9. Interpretation of SHAP force plot: analysis of performance in the full G2 game.**

**Table 11. Impact of important variables and other variables on game outcome during the first two quarters of G2 game.**

| First two quarters | | |
|---|---|---|
| Indicators | Sample Feature Values | Shap Values |
| H2_FG% | 0.058 | 0.1 |
| H2_AST | 2 | 0.04 |
| H2_PF | -4 | 0.04 |
| H2_DRB | 3 | 0.04 |
| Other_indicators | | 0.026 |

and H2_DRB. The Nuggets' field goal percentage being 5.8% higher than the Heat's had the most significant impact, increasing the Shap Value (contribution to the predicted probability of victory) by 0.1 for the Nuggets. Additionally, more H2_AST (2), fewer H2_PF (-4), and more H2_DRB (3) each contributed 0.04 to the Shap Value. At this point, under the combined influence of multiple features, the Shap Value indicating the Nuggets' win probability in Game 2 rose to 0.35, exceeding the average predicted value of 0.104.

As shown in Fig 8 and Table 12, by the end of the third quarter of Game 2, the top four features impacting the game result, in order of Shap Values, were H3_FG%, H3_DRB, H3_3P%,

**Table 12. Impact of important variables and other variables on game outcome during the first three quarters of G2 game.**

| First three quarters | | |
|---|---|---|
| Indicators | Sample Feature Values | Shap Values |
| H3_FG% | 0.07 | 0.55 |
| H3_DRB | 7 | 0.27 |
| H3_3P% | -0.08 | -0.18 |
| H3_PF | -8 | 0.135 |
| Other_indicators | | -0.045 |

**Table 13. Impact of important variables and other variables on game outcome during the full G2 game.**

| Full game | | |
|---|---|---|
| Indicators | Sample Feature Values | Shap Values |
| Game_3P% | -0.09 | -1.53 |
| Game_FG% | 0.03 | 0.787 |
| Game_DRB | 6 | 0.612 |
| Game_TOV | 2 | -0.568 |
| Other_indicators | | -0.126 |

and H3_PF. The Nuggets' three-point field goal percentage being 8% lower than the Heat's had a negative effect, reducing the Shap Value (contribution to the predicted probability of victory) by 0.18 for the Nuggets. However, a higher H3_FG% (0.07), more H3_DRB (7), fewer H3_PF (-8), along with the influence of other indicators, increased the SHAP value indicating the Nuggets' win probability in Game 2 to 0.90, which is higher than the average predicted value of 0.17.

As shown in Fig 9 and Table 13, for the entire Game 2, the top four features impacting the game result, in order of Shap Values, were Game_3P%, Game_FG%, Game_DRB, and Game_-TOV. The Nuggets' three-point field goal percentage being 9% lower than the Heat's had the most significant impact, reducing the Shap Value (contribution to the predicted probability of victory) by 1.53 for the Nuggets. Additionally, the Nuggets had 2 more turnovers than the Heat, reducing the Shap Value by 0.568. Ultimately, under the combined influence of multiple positive and negative features, the Shap Value indicating the Nuggets' win probability in Game 2 was -0.59, lower than the average predicted value of 0.235.

Further analysis of the impact of the Miami Heat's 9.3% higher three-point shooting percentage compared to the Denver Nuggets on the game's outcome reveals that the Nuggets made 11 out of 28 three-point attempts (39.3%), while the Heat were more efficient, making 17 out of 35 attempts (48.6%). If, in the actual game scenario, the Nuggets could have limited the Heat's three-point shooting percentage through effective tactical coordination and defensive strategies, reducing it by 6%, the outcome prediction model would yield a new probability of victory. With this adjustment, the feature would have decreased the predicted winning probability by only 0.8544, and the comprehensive effect on the function $f(x)$ would be 0.8956, which is higher than the base value of 0.235. In this scenario, the Nuggets would have won Game 2, and with this victory, they could have potentially swept the Heat with a 4:0 series win, securing the championship title for the 2022–2023 season ahead of schedule.

## 7 Conclusion

To explore the practical application of artificial intelligence technology in real-time prediction of basketball games, this study simulates the real-time acquisition of basketball game technical statistics. Based on the integration of XGBoost and SHAP models, a real-time prediction model for NBA game outcomes was established. The model conducted a quantitative analysis of the key factors influencing the game's outcome at different stages, overcoming the limitation of low explainability in traditional machine learning methods in this research area. It provides a reference for coaches' dynamic in-game decision-making and analysis of winning factors. On one hand, by utilizing XGBoost algorithm based on Bayesian optimization and grid search, a real-time prediction model for NBA game outcomes was constructed, demonstrating encouraging outcomes in predictive accuracy. As the game progresses and data is updated in real-time, the "real-time prediction" model offers higher reliability and precision compared to

the "pre-game prediction" model and has greater practical value than the "post-game prediction" model. On the other hand, the SHAP algorithm was employed to enhance the interpretability of the XGBoost prediction model, thereby analyzing the key factors influencing the game's outcome at different stages and quantifying and visualizing the results. The experiments indicated that field goal percentage, defensive rebounds, and turnovers are critical factors affecting the outcome across various stages of the game. Additionally, assists are key indicators of game outcomes during the first two quarters, while offensive rebounds and three-point shooting percentages become more influential as the game time decreases. This approach provides a new method and perspective for studying the winning factors of basketball events and can be used to explore the factors affecting the outcomes of competitive sports, constructing comprehensive and scientific game outcome prediction models tailored to the characteristics of different competitive sports.

Currently, this model only utilizes team technical indicator data for "real-time prediction" and "post-game prediction" of basketball games. To further enhance the performance and application value of the game outcome prediction model, future work will consider incorporating data on team tactics, player injuries, game time, and other relevant factors to predict and analyze the outcomes of basketball games. Additionally, to ascertain the generalizability and resilience of our prediction models, we plan to extend our methodological framework to other sports domains, including Hockey and Football. This expansion will potentially validate the versatility and robustness of our predictive approaches across varying sports disciplines.

## Acknowledgments

Thanks to all authors for their contributions.

## Author Contributions

**Conceptualization:** Yan Ouyang, Xuewei Li, Wenjia Zhou.

**Data curation:** Yan Ouyang, Xuewei Li, Wenjia Zhou.

**Formal analysis:** Yan Ouyang, Xuewei Li, Wenjia Zhou.

**Funding acquisition:** Xuewei Li, Liming Peng.

**Investigation:** Yan Ouyang, Xuewei Li, Wenjia Zhou, Wei Hong, Weitao Zheng, Feng Qi, Liming Peng.

**Methodology:** Yan Ouyang, Xuewei Li, Wenjia Zhou, Wei Hong.

**Project administration:** Liming Peng.

**Resources:** Yan Ouyang, Wei Hong.

**Software:** Yan Ouyang, Wei Hong.

**Supervision:** Weitao Zheng, Liming Peng.

**Validation:** Xuewei Li, Wenjia Zhou, Wei Hong.

**Visualization:** Wei Hong.

**Writing – original draft:** Yan Ouyang.

**Writing – review & editing:** Xuewei Li, Wenjia Zhou, Wei Hong, Weitao Zheng, Feng Qi, Liming Peng.

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
