## [Decision Letter · Decision Letter 0]

7 May 2024

PONE-D-24-12722Integration of machine learning XGBoost and SHAP models for NBA game outcome prediction and quantitative analysis methodologyPLOS ONE

Dear Dr. Peng,

Thank you for submitting your manuscript to PLOS ONE. After careful consideration, we feel that it has merit but does not fully meet PLOS ONE’s publication criteria as it currently stands. Your manuscript has been reviewed by two experts whose comments are appended below. Both reviewers are positive about your work but have raised minor issues that call for enhanced descriptions and motivations of your methods, along with implementing some robustness tests. Therefore, we invite you to submit a revised version of the manuscript that addresses the points raised during the review process.

We look forward to receiving your revised manuscript.

Kind regards,

Haroldo V. Ribeiro

Academic Editor

PLOS ONE

Journal Requirements:

   "This research was funded by the 14th Five-Year-Plan Advantageous and Characteristic Disciplines (Groups) of Colleges and Universities in Hubei Province (Grant Number: E Jiao Yan No. [2021] 5), Hubei Provincial Social Science Fund General Project “Research on Personalized Recommendation of Online Sports Education Resources Based on Knowledge Graph” (Grant Number: 2021330), and the Scientific and Technological Research Project of Hubei Provincial Education Department (Grant Number: B2021189)."

4. Please remove your figures from within your manuscript file, leaving only the individual TIFF/EPS image files, uploaded separately. These will be automatically included in the reviewers’ PDF.

Reviewers' comments:

Reviewer's Responses to Questions

**Comments to the Author**

1. Is the manuscript technically sound, and do the data support the conclusions?

Reviewer #1: Yes

Reviewer #2: Yes

2. Has the statistical analysis been performed appropriately and rigorously? 

Reviewer #1: Yes

Reviewer #2: No

3. Have the authors made all data underlying the findings in their manuscript fully available?

Reviewer #1: No

Reviewer #2: No

4. Is the manuscript presented in an intelligible fashion and written in standard English?

Reviewer #1: Yes

Reviewer #2: No

5. Review Comments to the Author

Reviewer #1: Report on "Integration of machine learning XGBoost and SHAP models for NBA game outcome prediction and quantitative analysis methodology"

Paper Description:

I have reviewed the article titled "Integration of machine learning XGBoost and SHAP models for NBA game outcome prediction and quantitative analysis methodology", submitted for publication in PLOS ONE. In this work, the authors investigate the possibility of real time prediction (and post-event prediction, as well) of basketball games belonging to the NBA. To achieve this, the authors train various machine learning algorithms, including kNN, SVM, LightGBM, Random Forest, Logistic Regression, Decision Tree, and XGBoost.

They utilize eleven game statistics (such as difference in field goal percentage, difference in free throw percentage and others) to train models for predicting home team win/loss with game data from the second (or third) quarter or full game data. Overall, the authors find that XGBoost produces the best-performing models, reaching close to 80% accuracy in predicting game results for models trained on third-quarter data and over 90% accuracy for post-game data. Additionally, they conduct a feature relevance analysis of the XGBoost models using SHAP values. Finally, they discuss their results extensively and apply their machine learning framework to analyze Game 2 of the NBA Finals in the 2022-2023 season.

Before recommending this paper for publication, I only wish the authors to address some minor issues.

General Comments/Issues:

- In the last paragraph of the Introduction, do the authors mean "Section" where it is written "Chapter"?

- In line 215, the authors mention removing outliers from their data. What characterizes an outlier in the context of basketball games, and why were they removed?

- Despite finding this paper to be generally well-written, Section 3.1, which describes the XGBoost algorithm, needs significant improvement. For instance, errors in summation indices can be easily found in Eqs. 1 and 2, and lines 180-181 are difficult to understand. The description of the algorithm is neither intuitive nor fluid in the current version, but it could be improved with minor adjustments.

Spelling Issue:

Line 375: ". the protection" -> ". The protection"

Reviewer #2: The manuscript is well-written, and the topic is also interesting. The data-driven analysis using ML tools is also rigorous. I have a few comments that the author should consider in a revised manuscript. Please check the comments uploaded as an attachment.

6. PLOS authors have the option to publish the peer review history of their article (what does this mean?). If published, this will include your full peer review and any attached files.

Reviewer #1: No

Reviewer #2: No

---

## [Author Response · Author response to Decision Letter 0]

21 Jun 2024

Response to Reviewers

We would like to resubmit our revised manuscript for consideration in PLOS ONE. The manuscript is entitled “Integration of machine learning XGBoost and SHAP models for NBA game outcome prediction and quantitative analysis methodology” PONE-D-24-12722.

The authors deeply appreciate the very detailed review of the manuscript and all the thoughtful comments that will undoubtedly help to improve it substantially. We have studied comments carefully and have made corrections which we hope meet with approval. All changes have been highlighted (dark red colour) with track changes in the revised manuscript. The following is a point-by-point response to the reviewers’ comments. We hope that our manuscript can now receive your recognition and approval.

Comments from the editors and reviewers:

Dear editors/referees, many thanks for your constructive and valuable criticisms. Our responses are presented below and we are looking forward and ready to respond to any future comments.

Journal Requirements:

Response: Thank you reviewers for listing the documents leading to the correction of the style requirements.

Response: Thank you for your valuable feedback and for outlining the guidelines on code sharing. In accordance with PLOS ONE's requirement for transparency and reproducibility, we have made all the author-generated code available on GitHub. You can access the code using the following link: https://github.com/YanOuyang514/NBA-game-outcome-prediction-and-quantitative-analysis-methodology.git.

 "This research was funded by the 14th Five-Year-Plan Advantageous and Characteristic Disciplines (Groups) of Colleges and Universities in Hubei Province (Grant Number: E Jiao Yan No. [2021] 5), Hubei Provincial Social Science Fund General Project “Research on Personalized Recommendation of Online Sports Education Resources Based on Knowledge Graph” (Grant Number: 2021330), and the Scientific and Technological Research Project of Hubei Provincial Education Department (Grant Number: B2021189)."

Response: Thank you for advising on PLOS ONE’s requirements for financial disclosure. We have amended the financial disclosure section in the cover letter as follows:

“This research was funded by the 14th Five-Year-Plan Advantageous and Characteristic Disciplines (Groups) of Colleges and Universities in Hubei Province (Grant Number: E Jiao Yan No. [2021] 5). The funder (Liming Peng) participated in the revision of the manuscript and in the decision to publish. 

Hubei Provincial Social Science Fund General Project “Research on Personalized Recommendation of Online Sports Education Resources Based on Knowledge Graph” (Grant Number: 2021330), and the Scientific and Technological Research Project of Hubei Provincial Education Department (Grant Number: B2021189). The funder (Xuewei Li) had participated in study design, data collection, and analysis, the decision to publish, as well as preparation of the manuscript.”

4. Please remove your figures from within your manuscript file, leaving only the individual TIFF/EPS image files, uploaded separately. These will be automatically included in the reviewers’ PDF.

Response: Thank you for your insightful comment. We remove and checked the figures in the manuscript to ensure that it adhered to the journal’s guidelines.

Response: Thank you for your valuable feedback. In response to your comments on the reference list, I have made the following adjustments:

-To support the critical roles of blocks and steals in affecting game outcomes, we have moved the original reference [41] to [29], and added references [30] and [31] to further substantiate this point.

-During the adjustment of the article order, the sequence of the original reference [40] (now shown as [43]) was inadvertently moved forward, which explains the discrepancy.

-We have corrected the citation formats for several references. Specifically, references [5], [6], [14], and [46] are in Chinese, so we prefixed the DOI link with "Chinese" to distinguish them from the English references. Similarly, reference [20] is in Turkish, so we prefixed the DOI link with "Turkish". Additionally, we addressed the capitalization issues in the journal names for references [8], [9], and [17]. Although reference [16] has not been retracted, its DOI link is no longer valid, so we removed the DOI link. Similarly, the DOI link for reference [29] is also invalid; however, we found the download link on the official website and replaced the DOI with this link.

We appreciate your thorough review and hope these changes meet your expectations.

Reviewers' comments:

Reviewer's Responses to Questions

Comments to the Author

1. Is the manuscript technically sound, and do the data support the conclusions?

Reviewer #1: Yes

Reviewer #2: Yes

2. Has the statistical analysis been performed appropriately and rigorously?

Reviewer #1: Yes

Reviewer #2: No

3. Have the authors made all data underlying the findings in their manuscript fully available?

Reviewer #1: No

Reviewer #2: No

4. Is the manuscript presented in an intelligible fashion and written in standard English?

Reviewer #1: Yes

Reviewer #2: No

5. Review Comments to the Author

Reviewer #1: Report on "Integration of machine learning XGBoost and SHAP models for NBA game outcome prediction and quantitative analysis methodology"

Paper Description:

I have reviewed the article titled "Integration of machine learning XGBoost and SHAP models for NBA game outcome prediction and quantitative analysis methodology", submitted for publication in PLOS ONE. In this work, the authors investigate the possibility of real time prediction (and post-event prediction, as well) of basketball games belonging to the NBA. To achieve this, the authors train various machine learning algorithms, including kNN, SVM, LightGBM, Random Forest, Logistic Regression, Decision Tree, and XGBoost.

They utilize eleven game statistics (such as difference in field goal percentage, difference in free throw percentage and others) to train models for predicting home team win/loss with game data from the second (or third) quarter or full game data. Overall, the authors find that XGBoost produces the best-performing models, reaching close to 80% accuracy in predicting game results for models trained on third-quarter data and over 90% accuracy for post-game data. Additionally, they conduct a feature relevance analysis of the XGBoost models using SHAP values. Finally, they discuss their results extensively and apply their machine learning framework to analyze Game 2 of the NBA Finals in the 2022-2023 season.

Before recommending this paper for publication, I only wish the authors to address some minor issues.

Response: We greatly appreciate the time and effort you put into our manuscript, as well as your valuable comments.

General Comments/Issues:

- In the last paragraph of the Introduction, do the authors mean "Section" where it is written "Chapter"?

Response: Thank you for pointing out this oversight. We agree that "Section" is the appropriate term instead of "Chapter" and have promptly corrected this error in the manuscript (Please see line 70-76). 

- In line 215, the authors mention removing outliers from their data. What characterizes an outlier in the context of basketball games, and why were they removed?

Response: Thank you for your question regarding our methods for outlier removal in the dataset of NBA games. I appreciate the opportunity to clarify this part of our study.

In the context of our research, an 'outlier' refers to data points that do not accurately reflect the typical conditions or performance indicators of an NBA game with regards to the objectives of our analysis. Specifically, we identified outliers as games that fell into two categories: 1) preseason games and All-Star games, which do not represent the standard competitive environment due to differing player lineups and competitive intents; and 2) games for which data was compromised due to bugs within our web scraping tool, leading to incomplete or inaccurate records.

Preseason and All-Star games were removed because these events do not reflect the actual competitive dynamics of regular season or playoff games. Preseason games often involve experimental lineups and varied player usage that differ significantly from regular season strategies, while All-Star games are exhibition matchups that prioritize entertainment and are not indicative of a team’s true capabilities or strategic approaches.

Furthermore, during the data validation process, we identified and excluded a small subset of game records where the collected statistics were incorrect due to technical anomalies in the data scraping process.

Thus, to ensure the integrity of our analysis which aims at modeling and predicting outcomes based on data reflective of regular NBA competitive play, these outlier games were removed from the dataset. This resulted in a final count of 3,710 valid games for the 2021–2023 seasons, providing a robust foundation for our predictive models.

We recognize the importance of transparency regarding data handling practices in research and have amended our manuscript to include a more detailed explanation of our process for identifying and excluding outliers, as you suggested. This revision provides readers with important context about the dataset composition and supports the reliability of our findings (Please see line 217-218).

Once again, thank you for your insightful feedback, which has helped improve the quality and accuracy of our research.

- Despite finding this paper to be generally well-written, Section 3.1, which describes the XGBoost algorithm, needs significant improvement. For instance, errors in summation indices can be easily found in Eqs. 1 and 2, and lines 180-181 are difficult to understand. The description of the algorithm is neither intuitive nor fluid in the current version, but it could be improved with minor adjustments.

Response: Thank you for your insightful feedback on our manuscript. We have carefully addressed your concerns by correcting the errors in the summation indices in Eqs. 1 and 2, and we have revised the description of the XGBoost algorithm to enhance its clarity and flow (Please see line 153-184). We believe these adjustments have substantially improved the intuitiveness and readability of Section 3.1. We are grateful for your valuable suggestions and hope that the revised manuscript meets your expectations.

Spelling Issue:

Line 375: ". the protection" -> ". The protection"

Response: Thank you for pointing out the error. We have corrected the spelling issue by changing "the protection" to "The protection" to ensure proper capitalization.

Reviewer #2: The manuscript is well-written, and the topic is also interesting. The data-driven analysis using ML tools is also rigorous. I have a few comments that the author should consider in a revised manuscript. Please check the comments uploaded as an attachment.

Response: Thank you very much for your encouraging comments and we are pleased to respond. Thank you for recommending our manuscript.

1.The authors primarily focus on the Basketball dataset. However, I feel the methods should be generalizable to other sports like Hockey and Football.

Response: We appreciate your suggestion and agree that generalizing the methodology to other sports can enhance the study’s scope and applicability. This point will be included as a future research direction in the conclusion section (Please see line 556-559).

2.Figure 6 is not at all clear. So what are the feature importance of the feature variables.

Response: Thank you for your insightful feedback regarding Figure 6. In response to your suggestion, we have added a supplementary table to enhance the clarity and understanding of the feature importance of the feature variables (Please see Tables 11-13). Additionally, we have revised certain textual descriptions to better articulate the importance of these features (Please see line 476-515). We appreciate your detailed review and trust that these modifications will address your concerns.

3.In your work, you have basically compared the various ML algorithms and based on a certain metric, the xGBoost is chosen. This is what you do in a Hackathon. But you need to tell a story with the Data. I would appreciate it if you first conducted an exploratory analysis of the data with some interesting plots. You have given descriptive statistics, but that’s not enough. You should also present the significance of the pairwise correlations.

Response: Thank you for the suggestion. We recalculated correlation coefficients and conducted significance tests, then replotted the correlation heatmap. Based on these results, we perfo

---

## [Decision Letter · Decision Letter 1]

8 Jul 2024

Integration of machine learning XGBoost and SHAP models for NBA game outcome prediction and quantitative analysis methodology

PONE-D-24-12722R1

Dear Dr. Peng,

We’re pleased to inform you that your manuscript has been judged scientifically suitable for publication and will be formally accepted for publication once it meets all outstanding technical requirements.

Kind regards,

Haroldo V. Ribeiro

Academic Editor

PLOS ONE

Reviewers' comments:

Reviewer's Responses to Questions

**Comments to the Author**

1. If the authors have adequately addressed your comments raised in a previous round of review and you feel that this manuscript is now acceptable for publication, you may indicate that here to bypass the “Comments to the Author” section, enter your conflict of interest statement in the “Confidential to Editor” section, and submit your "Accept" recommendation.

Reviewer #1: All comments have been addressed

Reviewer #2: All comments have been addressed

2. Is the manuscript technically sound, and do the data support the conclusions?

Reviewer #1: Yes

Reviewer #2: Yes

3. Has the statistical analysis been performed appropriately and rigorously? 

Reviewer #1: Yes

Reviewer #2: Yes

4. Have the authors made all data underlying the findings in their manuscript fully available?

Reviewer #1: Yes

Reviewer #2: No

5. Is the manuscript presented in an intelligible fashion and written in standard English?

Reviewer #1: Yes

Reviewer #2: Yes

6. Review Comments to the Author

Reviewer #1: All my previous comments were addressed and I am ready to recommend the publication of this work as it stands after the first round of revision.

Reviewer #2: The authors have significantly improved upon the manuscript and incorporated the changes and explanations I had asked for. I don't have any further comments.

7. PLOS authors have the option to publish the peer review history of their article (what does this mean?). If published, this will include your full peer review and any attached files.

Reviewer #1: No

Reviewer #2: No

---

## [Editor Report · Acceptance letter]

12 Jul 2024

PONE-D-24-12722R1 

PLOS ONE

Dear Dr. Peng, 

I'm pleased to inform you that your manuscript has been deemed suitable for publication in PLOS ONE. Congratulations! Your manuscript is now being handed over to our production team.

Kind regards, 

on behalf of

Dr. Haroldo V. Ribeiro 

Academic Editor

PLOS ONE